# Protein Content and Amino Acid Profiles of Selected Edible Insect Species from the Democratic Republic of Congo Relevant for Transboundary Trade across Africa

**DOI:** 10.3390/insects13110994

**Published:** 2022-10-29

**Authors:** Papy Nsevolo Miankeba, Alabi Taofic, Nkoba Kiatoko, Kambashi Mutiaka, Frédéric Francis, Rudy Caparros Megido

**Affiliations:** 1Faculté des Sciences Agronomiques, Université Pédagogique Nationale (UPN), Kinshasa 8815, Democratic Republic of the Congo; 2Faculté des Sciences Agronomiques, Université de Kinshasa (UNIKIN), Kinshasa 15373, Democratic Republic of the Congo; 3Unité d’Entomologie Fonctionnelle et Evolutive, Gembloux Agro-Bio Tech (ULiège), 5030 Gembloux, Belgium; 4International Centre of Insect Physiology and Ecology (ICIPE), Nairobi P.O. Box 30772-00100, Kenya

**Keywords:** entomophagy, food security, high-quality protein, clusterization, nutritional health

## Abstract

**Simple Summary:**

Many edible insect species are consumed in Africa, but their nutrient composition—taking into account processing methods used to increase their shelf life—is under-documented. To fill knowledge gaps concerning relevant species for cross-border trade throughout Africa, this study analyzed the protein content and the amino acid (AA) profiles of six commercially available species in the Democratic Republic of Congo (DRC). The protein content of the orthopteran and lepidopteran representatives is relatively comparable with values reported for meat, fish, and poultry in the FAO’s food composition database. Some species such as *Imbrasia ertli* (Lepidoptera) contained high values in essential AAs, supporting the use of edible insects for dietary supplementation for vulnerable populations with cereal-based diets. Furthermore, the study also reported that these insect species could be grouped in three clusters based on their AA profiles, since the AA profiles varied according to insects’ taxa. Representatives of the family Notodontidae contained both the lowest values in several AAs and the essential amino acid index (which is a rapid calculation to determine protein nutritional quality) as compared to Saturniidae and Gryllidae. Overall, our findings supported edible insects as nutrient-rich food and we call for enhancing cross-border trade of species linked to potential economic, social, and ecological benefits.

**Abstract:**

This study analyzed the protein content of ten edible insect species (using the Dumas method), then focused on the amino acid (AA) profiles of the six major commercially relevant species using HPLC (high-pressure (or performance) liquid chromatography). The protein contents varied significantly from 46.1% to 52.9% (dry matter); the Orthoptera representative yielding both the highest protein content and the highest values in three essential amino acids (EAAs). Regarding Lepidoptera species, the protein content of Saturniidae varied more than for Notodontidae. *Imbrasia ertli* gave the best example of a species that could be suggested for dietary supplementation of cereal-based diets, as the sample contained the highest values in five EAAs and for the EAA index. Furthermore, first-limiting AAs in the selected insects have also been pointed out (based on a species-specific AA score), supporting that the real benefit from eating insects is correlated to a varied diet. Additionally, preliminary insights into AA distribution patterns according to taxa provided three clusters based on protein quality and should be completed further to help tailor prescriptions of dietary diets. Since the AA composition of the selected insects was close to the FAO/WHO EAA requirement pattern for preschool children and met the requirements of 40% EAAs with high ratio EAAs/NEAAs, the current study endorses reports of edible insects as nutrient-rich and sustainable protein sources.

## 1. Introduction

Since 2019, new challenges driven by the pandemic outbreak have been unavoidably imposed on global food systems [1,2]. Taking these varied challenges and a global population expected to reach 9.8 billion people by 2050 into account [3,4], the problem of food security is likely to worsen worldwide more than previously predicted. With particular reference to developing countries in Africa facing numerous challenges (viz., fragility of food systems, overcrowding of cities, ineffective policies, and poverty) [2,5,6], sustainable and nutrient-rich foods should consequently be considered in order to mitigate food insecurity. Data on nutrient-rich foods are also required, all the more so as the focus is gradually shifting from the fatal effects of COVID-19 to the threat it poses to the production and daily supply of food, especially for children, smallholders, and vulnerable populations [7,8].

As one of the prevailing food cultures traditionally ingrained in most of Sub-Saharan African (SSA) countries, entomophagy (i.e., edible insect consumption by humans) should be meaningfully considered as a promising route for alleviation of food insecurity in several SSA countries [9]. This is in line with an increasing number of reports highlighting edible insects as valuable sources of essential proteins, amino acids (AAs), fatty acids, carbohydrates, vitamins, minerals, and other bio-functional compounds [10,11]. As a matter of fact, based on the history, custom, consumption habits, and status of edible insects in several SSA countries, it has been proven that edible insects could well match needs for human health [12,13,14,15,16].

Moreover, carefully considering insects as food could also contribute to efficient expansion of redundancy or diversity within food systems and to an increased resiliency to food shocks [7,17]. Such an endorsement is supported by a growing number of authors reporting significant diversity of edible insect species in many SSA countries, including the Democratic Republic of the Congo (DRC) (148 species), the Central African (CA) Republic (up to 96 species), Gabon (75 species), and Cameroon (31 species) [18,19,20,21,22]. 

Furthermore, similar patterns are reported throughout SSA countries in the use of some insect species as food, in harvesting techniques (mainly traditional), and/or in processing methods (e.g., roasting, boiling, or frying) of insects harvested from the wild. For example, species like *Imbrasia ertli* Rebel 1904 (Saturniidae, Lepidoptera) are widely consumed in Zambia, South Africa, Cameroon, Congo-Brazzaville, the CA Republic, Zimbabwe, Botswana, Angola, and the DRC. An additional example is given with *Carebara vidua* Smith 1858 (Formicidae, Hymenoptera) whose consumption is reported in more than 10 countries on the continent [22,23]. Such a finding should be pointed out for SSA countries given the need to develop sustainable solutions for mitigating food insecurity and increasing resiliency of food systems (at local and national scales) [7,8]. This should also be highlighted for enhancing cross-border trade of edible insects. Such an enhancement might boost local production of species linked to potential economic, social, and ecological benefits for rural and vulnerable communities involved in the insect value chain.

Sadly, the potential of African edible insect species as food and profitable foodstuffs has a long way to go. This assertion is supported by existing scientific knowledge pointing out gaps to be filled in most SSA countries regarding this practice, resting on lingering traditions. Based on some of these reports [24,25,26], food safety and nutritional composition issues of edible insects (taking into account the different traditional techniques used to process them prior to consumption, or to increase their shelf life) are two of the major concerns in order to gain society’s approval of insects as food [27,28]. Moreover, outcomes from investigations linked to the latest aforementioned concern might also enhance strategies ensuring nutrient-rich foods in the event of further food shocks.

As a relevant corpus of knowledge is necessary in SSA countries to reliably assess edible insect contribution to food security, the current study gives preliminary insights into the protein content of selected edible insects from the DRC and then focuses on the AA profiles of major commercially relevant species that might be targeted for cross-border trade with SSA countries. Its results might contribute (along with existing nutrient data for African edible insect species) to building a valuable baseline that could help to develop nutritious diets for vulnerable populations still suffering from food insecurity and malnutrition.

## 2. Materials and Methods

### 2.1. Sample Collection and Identification

Ten edible species from two orders of insects (Table 1) were randomly collected from different markets in Kinshasa (the capital city of the DRC) and transferred to the Agro-Bio Tech entomological laboratory at Gembloux (University of Liège, Belgium). To avoid degradation during storage or transportation, insect samples were stored as purchased in 50 mL Falcon tubes filled with one volume of norvanol (88.5% *v*/*v* ethanol, 2.7% *v*/*v* diethyl ether, 8.6% *w*/*v* water), then kept cool at an average temperature of 3 ± 0.5 °C. The identification process was carried out based on dichotomous keys [29] and comparisons with reference specimens kept at the Gembloux Entomological Conservatory [30]. Insects that could not be taxonomically identified due to degradation during the processing phases (i.e., sun-drying or smoking) (Table 1) were simply enlisted using their ethnospecies—that is, corresponding vernacular names as used in local markets—and cross-checked with recent literature reviewing edible insect species from the DRC [22].

### 2.2. Protein Content Analysis

The protein content of the ten samples (Table 1) was analyzed following the Dumas method according to AOAC (Association of Official Analytical Chemists) method 968.06 [31]. The samples were freeze-dried for 48 h and stored at −18 °C. Prior to analysis (performed using a RapidN cube [V4.0.6] from Elementar Americas, Inc., Ronkonkoma, NY, USA), the system was calibrated, aspartic acid being used as the nitrogen calibration standard according to the manufacturer’s protocol. The whole bodies of sampled insects were ground up to be powdered. Approximately 200 mg of each sample was weighed, wrapped, and tightly pelleted in nitrogen-free paper. Afterward, three replications of each were used to determine whether the nitrogen values obtained were acceptable based on the known nitrogen content of the aspartic acid, and from satisfactory results, the protein content was calculated using the conversion factor of 4.76 [32,33,34].

### 2.3. Total and Sulfur Amino Acid Analysis

The six major commercially relevant species—based on reports by Nsevolo et al. [22] and Nsevolo et al. [35]—were selected (out of the ten sampled insect species), then ground to pass through a 750 μM sieve before the analysis process. Amino acid profiles were determined at the chemistry laboratory of Gembloux Agro-Bio Tech (University of Liège, Belgium) based on previously described methods [36]. In brief, each of the six samples (5.0 mg of protein) was acid hydrolyzed (with 10 mL of 6N HCl containing phenol at 0.1%) in vacuum-sealed hydrolysis (Schott glassware) at 110 °C for 24 h. After hydrolysis, the opened Schott brand bottles (100 mL) were cooled during addition of NaOH 7.5N to adjust pH to 1, and then with NaOH 1N to end pH at 2.2 [36]. Norleucine in citrate buffer was added to the HCl as an internal standard. The final solution was filtered using a 0.45 μM polytetrafluorethylene syringe filter, and amino acid composition was determined using an automated amino acid analyzer (HPLC, Biochrom 20 Plus, Biochrom Ltd., Cambridge, UK). A sodium high-performance protein hydrolysate column (200 × 4.6 mm, with a flow rate of 20 mL/h), commercial standards, sodium citrate buffers, and ninhydrin reagent were employed to achieve derivatization. Performic acid oxidation (16 h, 4 °C) prior to acid hydrolysis (24 h, 110 °C) permitted determining sulfur-containing amino acids like cysteine and methionine according to previously published methods (AOAC 994.12) [36]. As conventional acid hydrolysis is destructive for tryptophan, alkaline hydrolysis with barium hydroxide (20 h, 110 °C) was performed to quantify tryptophan based on a previously described technique (AOAC 988.15) [36]. Amino acid content (expressed as g/100 g dry sample) was calculated based on the peak area for known concentration of amino acids.

**Table 1 insects-13-00994-t001:** Nomenclature of sampled edible insect species—[n/d] stands for “no data”.

Order	Family	Scientific Name ^(9)^	Vernacular Name ^(9)^	Number of Countries Reported	Selected Host (and/or Food) Plant Family Reported in the DRC
Lepidoptera	Saturniidae	Und. sp_1	Bingubala jaune ^(1) “S”^	n/d	n/d
		Und. sp_2	Binkubala ^(2)^	n/d	n/d
		*Cirina forda* Westwood 1849 ^(3)^	Mikwati ^”S”^	17	Anacardiaceae, Apocynaceae, Combretaceae, Euphorbiaceae, Fabaceae.
		*Imbrasia ertli* Rebel 1904 ^(4)^	Misati ^”D”^	9	Achariaceae, Apocynaceae, Dennstaedtiaceae, Euphorbiaceae, Fabaceae
		*Imbrasia rectilineata* Sonthonnax 1899 ^(5)^	Mangaya ^”D”^	7	Annonaceae, Fabaceae, Myrtaceae, Ochnaceae, Phyllanthaceae
		*Imbrasia* sp. ^(6)^	Makonzo ^”D”^	±7	
	Notodontidae	Und. sp_3	Mifwangi fwangi ^”D”^	n/d	n/d
		*Elaphrodes lactea* Gaede 1932 ^(7)^	Tunkubi ^”D”^	3	Apocynaceae, Burseraceae, Combretaceae, Dennstaedtiaceae, Dipterocarpaceae, Fabaceae, Hypericaceae, Loganiaceae, Ochnaceae, Rubiaceae
		*Elaphrodes* sp. ^(8)^	Mingingi ^”D”^	±3	
Orthoptera	Gryllidae	Und. sp_4	Makonki ^”D”^	±11	Field crops, vegetables

^(1)^ This ethnospecies could refer to *Gonimbrasia zambesina* Walker 1865, to *Imbrasia rectilineata* Sonthonnax 1899, to *Bunaeopsis aurantiaca* Rothschild 1895, or to *Athletes semialba* Sonthonnax 1904 (R. M. Lundanda, personal communication, 15 May 2022). ^(2)^ This ethnospecies could indistinctly refer to *Cinabra hyperbius* Westwood 1881 or to *Lobobunaea saturnus* Fabricius 1793 [22,37]. ^(3)^ This species is one of the major insects reported as food throughout Africa, under different local names according to Latham et al. [37], including “Amacimbi” (in Ndebele; Zimbabwe), “Ilir” (in Bekwel; Congo-Brazzaville), “Fikoso” (in Bemba, Lala-Bisa; Zambia), “Kadwisa” (in Nyanja; South Africa), “Minlone” (Ewondo; Cameroon), “Mpampala” (Western Téké; Gabon), “Nato” (Setswana; Botswana), “Ndinguiza” (Aka; Central African [CA] Republic), and “Nkuati” (Kongo; Angola). The caterpillar is also consumed in Burkina Faso, Nigeria, Mozambique, Namibia, Ghana, Togo, and Chad [23]. ^(4)^ This species is used as food throughout Central and Southern Africa under different local names, including “Avamukundu” (Shona; Zimbabwe), “Cuva” (Mbunda; Zambia), “Ovungu” (Umbundu; Angola), and “Makalampapa” (Chewa; Malawi) [37]. The caterpillar is also consumed in South Africa, Cameroon, CA Republic, Congo-Brazzaville, and Botswana [23]. ^(5)^ The species also known as *Gonimbrasia richelmanni* (Weymer, 1909) [22] is distributed and/or consumed across African countries including Congo-Brazzaville (locally called “Binkélé; Lari), Tanzania (likely called “Insega”; Malila), Angola, Uganda, Kenya, Malawi, and Zambia [37]. ^(6)^ The *Imbrasia* (*Nudaurelia*) genus has been reported as the most represented taxa of edible insects in the DRC [22], with about 16 edible species consumed nationwide, including *I. alopia* Westwood 1849, *I. anthina* Karsch 1892, *I. dione* Fabricius 1793, *I. epimethea* Drury 1773, *I. obscura* Butler 1878, *I. rubra* Bouvier 1920, *I. truncata* Aurivillius 1909, *I. wahlbergi* Boisduval 1847, *I. anthinoides* Rougeot 1978, *I. macrothyris* Rothschild 1906, *I. rhodina* Rothschild 1907, *I. eblis* Strecker 1876, and *I. oyemensis* Rougeot 1955. Regretfully, the exact match between the reported vernacular name (ethnospecies) and the corresponding Linnaean species could not be found. ^(7)^ This species is also consumed in a certain number of African countries, including Congo-Brazzaville (locally called “Shushu”), Zambia (called “Kakandu” or “Tunkubi”), and probably in Zimbabwe as well [37]. Furthermore, the species has been ranked the top polyphagous lepidopteran consumed in the DRC, as it feeds on 30 different plant species countrywide [22]. ^(8)^ For the *Elaphrodes* genus, 3 other species exist apart from *E. lactea*, namely: *E. duplex* (Gaede 1928), *E. nephocrossa* (Bethune-Baker 1909), and *E. simplex* (Viette 1955) [37]. *E. fusca* and *E. erato* cited by these authors are subspecies. These species need further studies given the pending questions as to which ones are used as food in Africa. ^(9)^ For scientific names, “Und. sp.” stands for “Undetermined species”. Vernacular names (V.N.) for lepidopterans refer to caterpillars, as they were sampled in their larval stage. The representative of Orthoptera was sampled as adult (imago). Superscript letters next to each V.N. are as follows: “S” = smoked; “D” = sun-dried. They indicate the processing methods of insects as they were purchased and stored.

### 2.4. Statistical Analysis

The protein content of the sampled edible insect species was compared to data reported in the FAO/INFOODS food composition table for Western Africa (Table 2) [38] for the common conventional animal-based protein sources in the DRC, according to reports from Nsevolo et al. [17]. In addition, statistical analysis of protein and amino acid contents (expressed as mean ± SD) in the edible insect samples were performed using R statistical software (version 3.6.1). Normalcy of data was checked using the Anderson–Darling test performed on Minitab (version 19.1.1) for Windows, whereas the R Commander GUI was used to complete analysis of variance (ANOVA) and Tukey’s test for comparison of means (with *p* values < 0.05). For further exploration (clusterization) of data and comparison with the AA profiles for selected insect species derived from the literature (Appendix A), correspondence analysis was carried out using the R statistical packages (FactoMiner and FactoShiny). 

## 3. Results

### 3.1. Protein Content

The protein content of the selected edible insect species is presented in Figure 1. As can be seen, the protein content of the nine lepidopterans varies from species to species, ranging between 40–50% (DM). In addition, the protein content of the three Notodontidae varied less as compared to the Saturniidae representatives (coefficients of variation = 1.47 and 8.01%, respectively). It should also be pointed out that high protein content (52.9%) was also recorded for the Orthoptera representative. Based on results from the statistical analysis, significant difference (F = 268.35; *p* < 0.001) was found among the sampled insects, the Gryllidae (Orthoptera) representative ranking at the top of the first three, followed by two representatives of Saturniidae (Lepidoptera)—namely *I. ertli* and Und. sp_1 (an unidentified species locally called “Bingubala jaune”). 

Furthermore, data on protein content (illustrated on Figure 1) have also been compared to the protein content of the most common animal-based foods in the DRC [17] from official nutrient composition tables (notably, the food composition tables available on the Food and Agriculture Organization website) [38]. Based on selected entries summarized in Table 2, the protein content of the sampled edible insect species was found to be relatively comparable to values reported in the FAO/INFOODS food composition table for Western Africa for the common conventional protein sources (poultry, fish, beef and pork) in the DRC. 

### 3.2. Amino Acid Profiles

Table 3 indicates the AA composition characterizing the protein quality of the major commercially relevant edible insect species in the DRC (namely: *C. forda*, *I. ertli*, “Binkubala”, *Elaphrodes* sp., “Mifwangi-fwangi”, and “Makonki”) (Table 1), based on reports from Nsevolo et al. [22] and Nsevolo et al. [35]. Although the content of essential or non-essential amino acids (NEAAs) varies between edible insect species, results showed that the six selected species contained 18 AAs, with essential amino acid (EAA) profiles close to the FAO’s recommended values (Figure 2).

Furthermore, *I. ertli* (Saturniidae) should be highlighted, as its contents of Ile, Phe, Thr, and Trp were the highest compared to the other species analyzed in this study. Moreover, *I. ertli* also had the highest Lys content along with another representative of Saturniidae (“Binkubala”—Und. sp_2) and the highest essential amino acid index (EAAI) (Table 3). In addition, it is noteworthy that the representative of Orthoptera (“Makonki”—Gryllidae) had the highest content of three EAAs (namely, Met, Val, and Leu) and the lowest value for Phe, whereas an unidentified representative of Lepidoptera (“Mifwangi fwangi”—Und. sp_3) showed both the lowest values for all the EAAs (except for Phe) (Figure 2) and the lowest EAAI (0.90) (Table 3). As for NEAAs, Gln and Asn predominated among the 18 AAs identified in the sampled edible insects. The representatives of Saturniidae (“Binkubala”—Und. sp_2) and Gryllidae (“Makonki”—Und. sp_4) yielded the highest values in Asn and Gln (respectively). 

Individual values for all EAAs (except for Trp) from the sampled insects were also compared to the EAA content of conventional foods of animal origin. Related results illustrated in Appendix A showed that the EAA profiles from sampled insects were comparable to common animal-based protein sources, taking human requirements for a healthy diet into account [39,40,41]. However, as can be seen in Table 4, the first limiting AAs for the selected insects (by computing the ratio of each amino acid compared to the reference profile) [41] were Leu for all the Lepidoptera species and sulphur-containing AAs (i.e., Met and Cys) for the Orthoptera representative (“Makonki”—Und. sp_4). Additionally, this latter species also had both the highest NEAA content (notably in Arg, Gln, Gly, Ala, and Pro) and AA score (0.85).

**Table 3 insects-13-00994-t003:** AA composition (g/100 g DM) of the main commercial edible insect species.

	Saturniidae			Notodontidae		Gryllidae
Amino acids (AAs) ^a^	*Cirina forda*	*Imbrasia ertli*	“Binkubala”	*Elaphrodes* sp.	“Mifwangi fwangi”	“Makonki”
Essential AAs (EAAs)						
Valine (Val)	3.61 ± 0.093	4.12 ± 0.024	3.85 ± 0.103	3.31 ± 0.165	3.20 ± 0.067 ^#^	4.28 ± 0.083 *
Isoleucine (Ile)	2.69 ± 0.065	3.00 ± 0.034 *	2.82 ± 0.076	2.32 ± 0.069	2.29 ± 0.081 ^#^	2.98 ± 0.036
Leucine (Leu)	3.85 ± 0.105	4.30 ± 0.055	4.10 ± 0.110	3.70 ± 0.092	3.63 ± 0.084 ^#^	5.40 ± 0.093 *
Lysine (Lys)	4.39 ± 0.116	4.91 ± 0.035 *	4.91 ± 0.132 *	4.07 ± 0.090	3.62 ± 0.006 ^#^	4.34 ± 0.012
Threonine (Thr)	3.05 ± 0.000	3.42 ± 0.014 *	3.38 ± 0.091	2.59 ± 0.030	2.50 ± 0.050 ^#^	2.76 ± 0.004
Phenylalanine (Phe)	2.75 ± 0.034	3.17 ± 0.153 *	2.93 ± 0.078	2.57 ± 0.065	2.35 ± 0.079	2.26 ± 0.048 ^#^
Methionine (Met)	0.95 ± 0.008	1.15 ± 0.011	1.08 ± 0.029	1.05 ± 0.006	0.82 ± 0.093 ^#^	1.17 ± 0.036 *
Histidine (His)	2.24 ± 0.008	2.19 ± 0.025	2.58 ± 0.069 *	1.95 ± 0.040	1.29 ± 0.040 ^#^	2.08 ± 0.059
Tryptophan (Trp)	0.71 ± 0.288	1.01 ± 0.027 *	0.77 ± 0.021	0.99 ± 0.150	0.53 ± 0.010 ^#^	0.73 ± 0.012
EAAs	24.24	27.27	26.42	22.53	20.23	26.01
Non-essential AAs (NEAAs)						
Tyrosine (Tyr) ^b^	3.55 ± 0.046	4.04 ± 0.071	4.23 ± 0.113 *	3.62 ± 0.041	3.29 ± 0.107 ^#^	3.46 ± 0.086
Arginine (Arg)	3.30 ± 0.110	3.56 ± 0.085	3.54 ± 0.095	3.12 ± 0.054	2.88 ± 0.101 ^#^	4.28 ± 0.142 *
Aspartic acid (Asn)	5.37 ± 0.045	5.87 ± 0.011	5.87 ± 0.157 *	4.92 ± 0.089	4.76 ± 0.081 ^#^	5.19 ± 0.008
Glutamic acid (Gln)	7.13 ± 0.081	7.65 ± 0.025	7.71 ± 0.207	6.63 ± 0.117	5.88 ± 0.161 ^#^	7.53 ± 0.064 *
Serine (Ser)	3.21 ± 0.036	3.72 ± 0.034 *	3.37 ± 0.09	2.62 ± 0.010	2.60 ± 0.077 ^#^	3.02 ± 0.036
Glycine (Gly)	2.82 ± 0.037	3.11 ± 0.021	3.04 ± 0.081	2.38 ± 0.026	2.22 ± 0.004 ^#^	3.78 ± 0.192 *
Alanine (Ala)	2.98 ± 0.063	3.24 ± 0.048	3.31 ± 0.089	2.90 ± 0.038 ^#^	2.90 ± 0.003 ^#^	5.63 ± 0.401 *
Cysteine (Cys)	1.01 ± 0.008	1.07 ± 0.028	1.14 ± 0.030 *	0.61 ± 0.014 ^#^	0.83 ± 0.022	0.69 ± 0.006
Proline (Pro)	2.87 ± 0.161	3.16 ± 0.137	2.86 ± 0.077	2.39 ± 0.085	2.26 ± 0.072 ^#^	3.97 ± 0.170 *
NEAAs	32.19	35.41	35.05	29.18	27.59	37.55
Total AAs	56.43	62.68	61.47	51.71	47.82	63.56
Ratio EAAs/NEAAs	0.75	0.77	0.75	0.77	0.73	0.69
EAAI ^c^	1.11	1.36	1.22	1.04	0.90	1.15

^a^ The amino acid (AA) abbreviations used here (and throughout the rest of the text) are in accordance with the International Union of Biochemistry and Molecular Biology to designate proteinogenic AAs [39]. Values are given as mean ± standard deviation from triplicate determinations. For each AA, ‘*’ indicates the highest value (i.e., max) whereas ‘^#^’ indicates the lowest one (i.e., min). ^b^ Conditional EAA. ^c^ Essential amino acid index calculated using human requirements based on the recommended protein pattern (individual AA in g/100 g protein, i.e., %) from FAO/WHO/UNU expert consultation [40]. According to Nielsen [41,42], values for methionine + cysteine and phenylalanine + tyrosine were used in calculation of the EAAI.

**Figure 2 insects-13-00994-f002:**
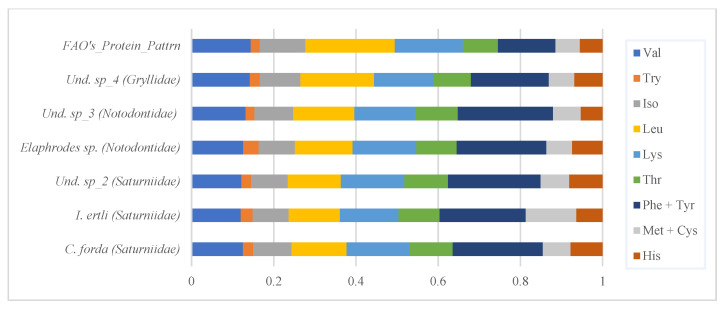
Comparison of EAA distribution in the protein fraction (% of total AAs) of six selected edible insects to protein pattern (individual AAs in g/100 g protein, i.e., %) based on recommendation by the FAO [40].

Finally, results of Table 3 were further explored through correspondence analysis (CA) to confirm whether the studied edible insect species showed significant differences based on their taxonomical membership (Figure 3). CA explained 87% of the variance with the first two principal dimensions and clearly suggested three distinct groups based on the EAA and NEAA composition of the studied insects. Cluster 1 (including *I. ertli*, *C. forda*, and “Binkubala”—Saturniidae) was associated with a higher content of Phe, Asn, Lys, His, and Ser; whereas cluster 2 (including Und. sp_4—Gryllidae) was characterized by a higher content of Ala, Leu, Arg, Pro, Gly, and Val. Conversely, cluster 3 (including *Elaphrodes* sp.-Notodontidae) was associated with the lowest content of five EAAs (namely Val, Ile, Leu, Lys, Thr, and His) and seven NEAAs (Table 3). 

**Table 4 insects-13-00994-t004:** EAA ratios calculated based on the FAO/WHO pattern for preschool children (2–5 years old), estimated AA score, and limiting AA of the selected insects.

	Edible Insects	Saturniidae		Notodontidae		Gryllidae	
EAAs		*C. forda*	*I. ertli*	“Binkubala”	*Elaphrodes* sp.	“Mifwangi fwangi”	“Makonki”
Valine		1.00	1.14	1.07	0.92	0.89	1.19
Tryptophan		1.10	1.58	1.20	1.54	0.83	1.15
Isoleucine		1.00	1.11	1.04	0.86	0.85	1.10
Leucine		0.71	0.80	0.76	0.68	0.67	1.00
Lysine		0.97	1.09	1.09	0.90	0.80	0.96
Threonine		1.33	1.49	1.47	1.13	1.09	1.20
Phenylalanine + Tyrosine		1.57	1.80	1.79	1.43	1.41	1.43
Methionine + Cysteine		0.89	1.93	1.01	0.75	0.75	0.85
Histidine		1.49	1.46	1.72	1.30	0.86	1.38
AA score ^a^		0.71	0.80	0.76	0.68	0.67	0.85
Limiting AA		Leu	Leu	Leu	Leu	Leu	Met + Cys

^a^ AA score = (mg of AA in 1 g of test protein/mg of AA in 1 g of reference protein) according to Nielsen [42], with preschool children’s EAA requirement pattern adapted from the FAO [40].

### 3.3. EAA Profiles Compared to Data Derived from Literature

Correspondence analysis (CA) was performed to compare the EAA profiles of the studied insect species with data (dry matter basis) on selected edible insect species derived from the literature (Appendix A). Related results (illustrated in Figure 4), suggest similar EAA profiles between the studied representative of Gryllidae (namely, Und. sp_4: “Makonki”) and the remaining representatives of Orthoptera (including *A. domesticus*, *Gryllodes sigillatus*, *Gryllus bimaculatus,* and *G. assimilis*). Likewise, CA also supports similar EAA profiles (*a*) between *C. forda* (“Mikwati”) and data derived from the literature for the same species, and (*b*) between *I. ertli* (“Misati”) and Und. sp_2 (“Binkubala”) with *I. obscura* from the literature (Appendix A). It is worth mentioning that the CA also suggests similar EAA profiles between some representatives of Lepidoptera (namely *I. truncata*, *I. epimethea*, and *N. oyemensis*) that contained higher levels of some EAAs (namely Lys, Met + Cys, and Phe + Tyr) than Orthopterans (Figure 4).

## 4. Discussion

Information regarding the nutritional composition of edible insects in SSA countries is limited, disparate in terms of methodology or data quality, and may vary depending on the location of the published sources [10,43,44,45]. However, available reports claim that edible insects are nutritious and should be seriously considered as a sustainable alternative to resource-intensive meat production [11,46,47]. Current results regarding the sampled edible insects from the DRC aligned with the aforementioned reports while complementing them in terms of protein content, AA profiles, and the chemical characterization of relevant edible insect species for cross-border trade among SSA countries.

Moreover, the current study supports reports that edible insects should be meaningfully considered in interventions aiming to alleviate food insecurity, or to prevent malnutrition and morbidity due to inadequate nutrient intake [10,27,46,47,48,49]. This assertion is based on significant protein content reported for the different species analyzed (46.0% on average for Lepidoptera representatives), as well as high-quality protein in terms of AA spectra and EAAI—which is a rapid calculation to determine protein nutritional quality [42]. Based on the results, all the analyzed species met the FAO/WHO requirements of 40% EAAs and exhibited a high ratio of EAAs/NEAAs (ranging from 0.69 to 0.77). Furthermore, on average, the concentrations of EAAs (especially Met + Cys, reported to be in low amounts in insect proteins) [10] were close to the FAO/WHO EAA requirement pattern for preschool children (2–5 years) and adults (>18 years) according to the FAO [40], with some exceptions, however (especially for Leu). 

The prime example of nutrient-rich edible insect species in this study was *I. ertli* (Saturniidae, Lepidoptera), as the species contained the highest values in four EAAs (namely Ile, Phe, Thr, and Trp) and the highest EEAI as compared to the remaining selected insects. In addition, the same species also contained high values of Lys, which is essential for healthy growth in children, for calcium absorption, and to form collagen for healthy connective tissue. In addition, Lys is also well-known to be the limiting AA in cereal proteins [50,51]. This is noteworthy as it gives a relevant example of an edible insect species, broadly consumed across Africa (including Zambia, South Africa, Cameroon, Congo-Brazzaville, the CA Republic, Zimbabwe, Botswana, Angola, and the DRC) [22,23,52], which could be used for dietary supplementation given that cereals are assumed as staple diets in many developing countries [43].

An additional example of a species with high levels of nutritive proteins was the representative of Orthoptera (“Makonki”—Gryllidae). Regretfully, the unidentified sample at species level (due to degradation during the processing phase to increase its shelf life) was simply enlisted using its related ethnospecies. Regardless of this challenge of accuracy in Linnaean identification (and species correspondence with vernacular names), the sample showed the highest protein content, the highest values in three EAAs (namely Val, Leu, and Met) and five NEAAs (namely Arg, Gln, Gly, Ala, and Pro). Taking into consideration the varied roles of the aforementioned AAs in human growth or health and their interactive networks in the body [39,53], these findings support an additional species that should be targeted for better description and chemical characterization. Mass production of such species should also increase access to proteins of animal origin. This could help alleviate nutrition insecurity and poor diets (which are incidentally an aggravating factor of the ongoing COVID-19 pandemic) [2]. 

Pragmatically speaking, the fortification and enrichment of food products with edible insect powder could be an effective solution to address food-related challenges and other public health issues linked to poor diets. This could be effectively implemented to mitigate food insecurity in developing countries in Africa (such as the DRC, Congo-Brazzaville, and Angola) where entomophagy is culturally accepted. Indeed, such an approach has been successfully implemented in Asia (e.g., Thailand) where food companies produce pasta enriched with 20% cricket powder [3]. This is also already considered in some SSA countries (notably Kenya) where the consumer acceptability and physical quality of bread and cookies enriched with 10% house cricket powder were comparable to control products [54]. 

However, it is worth mentioning that the AA score, which measures the content of the first limiting AA compared to requirements of preschool age children-as a part of the protein digestibility corrected amino acid score (PDCAAS) assay (the method of choice for dietary protein quality assessment) [41] indicated Leu as the limiting AA in all the selected lepidopterans, and sulphur-containing AAs (Met and Cys) for the Orthoptera representative (Table 4). Similar findings are also reported for a number of commercially relevant edible insect species worldwide [55,56]. 

Such reports on specific limiting AAs in edible insects should raise awareness that the real benefit one can derive from eating edible insects is necessarily correlated to a wholesome and varied diet (to optimally meet human requirements for essential nutrients as suggested by the FAO/WHO) [40]. Therefore, nutritious and attractive new products (or formulations), coupled with relevant information derived from a better knowledge of limiting AAs in edible insects, should be made available for commercially relevant insect species. This information could be made accessible to consumers through effective labeling of insect-based products (on a species-specific basis). Moreover, operational labeling of insect-based products and efficient outreach could encourage consumers toward edible insects as healthy foods and part of healthy diets. This could also help reduce reliance on meat and fast foods linked to the prevalence of chronic diet-related conditions (such as obesity, hypertension, and diabetes) emerging in developing African countries [3,57]. 

Furthermore, considering questions related to variation of protein content and AA spectra both within and between species from the three taxa analyzed, CA results suggested three clusters based on the AA profiles of the selected insect species. Membership in cluster 1 was positively associated with Saturniidae insect species, containing the highest values of three EAAs (Phe, Lys, and His) and two NEAAs (Asn and Ser). Membership in cluster 2 was associated with Orthoptera species, exhibiting both the highest content of two EAAs (Leu and Val) as well as four NEAAs (Ala, Arg, Pro, and Gly), and low values of aromatic AAs (principally Phe). Membership in cluster 3 was associated with Notodontidae representatives, showing the lowest values of six EAAs (Val, Ile, Leu, Lys, Thr, and His), as well as in all the NEAAs (except Cys). This finding is in part probably associated with the taxonomic group, since it is well known that closely related species use the same class of resources [58]. As some edible insect species may be less beneficial to human health than hitherto believed [44], complementary insights into distribution patterns of AAs (or their relative proportions according to the taxonomical origin of edible insect species) might be relevant for tailored prescriptions for diets in relation to the nutritive profiles of edible insects. 

Documenting the protein and AA richness of insect species used as food in the DRC was not without its challenges. First and foremost, as a number of methods exist to determine protein quality, comparisons with available data as derived from literature for the studied species were complex. For example, the protein content of *C. forda* (Saturniidae) whose caterpillar is consumed in no less than 17 countries throughout Africa, varied from 20.2–33.1% to 62.5–74.4%, making comparisons with current results (46.1%) tricky and inconclusive. A similar challenge has been documented by a couple of studies as well, notably Rumpold and Schlüter [10] and Payne et al. [44], who reported tremendous data variations on a dry matter basis (based on a compilation of 236 nutrient compositions of edible insects as published in the literature). Such variations might be related to the different measuring methods or could originate either from distinct development stages, from differences in insects’ diets (i.e., feed), ecology, or notably from processing methods. 

Notwithstanding, the reported protein content for Lepidoptera from the literature (45.38%) [10] was close to the average protein content of the studied representatives of Notodontidae and Saturniidae (Figure 1). In addition, the studied representative of *Imbrasia* showed similar content of some EAAs (namely Trp, Thr, Phe + Tyr, Met + Cys, Ile, and His) to its congeneric species *I. obscura* (with SDs as follows: 0.008, 0.370, 0.063, 0.367, 0.421, and 0.132, respectively) (Table 3 and Appendix A). Likewise, Und. sp_2 (“Binkubala”) yielded relatively similar EAA contents to the other representatives of Saturniidae (*I. obscura*, *I. ertli,* and *I. truncata*) (Table 3, Appendix A), supporting its membership in cluster 1 (Figure 3 and Figure 4). Similarly, the protein content of the studied representative of Gryllidae was relatively close to the value reported for orthopterans (61.32% on average) [10]—as was its EAA profile (Figure 4). 

Although the protein and AA contents of the studied edible insect species were in the range of data previously reported for representatives of orthopterans or lepidopterans (Appendix A) as derived from the literature [10,59,60,61,62,63,64,65,66,67], the current study supports the call for adherence to global standards of nutrient composition analysis to circumvent methodology-related variations hampering (at least in part) the full understanding of insect nutritional composition [44]. 

For a reliable assessment of the nutritional quality of insect proteins, this study also suggests complementary investigations of protein digestibility and anti-nutrient compounds present in the studied edible insects. The most commonly used approaches in evaluating protein or AA digestibility (and bioavailability) include AA score, biological value, net protein utilization, in vitro or in vivo protein digestibility, PDCAAS, and the digestible indispensable AA score (DIAAS) [40,41,68]. The latter method, recommended by the FAO, also incorporates up-to-date scientific information about AA reference patterns and AA digestibility, as well as the effects of anti-nutritional factors and food processing [40,68].

A further impediment that hinders rigorous assessment of the global contribution of insects to human nutrition is related to traditional ecological knowledge (TEK). As TEK is basically shared orally from one generation to another, the accurate taxonomical identification of many edible insect species using Linnaean nomenclature is tricky. In effect, as reported in Table 1, too many ethnospecies used as food in rural communities (and whose common names may vary according to the plants they feed on, their life stage, or the local languages used) are under-documented. This issue is already reported for edible insect species from the DRC [69,70] and for many other edible insect species harvested from the wild and traditionally processed by rural communities in most SSA countries [71,72]. Accordingly, a call is made for considering citizen science [69] as a pathway to unravel insect species amidst vernacular names used for edible insects in the DRC, or from the tremendous number of ethnospecies still hitherto unidentified [22].

## 5. Conclusions and Perspectives

This study provided insights into the protein content and AA spectra of selected commercial edible insects from three taxa and showed the high quality of insect proteins (as reflected in the various protein quality indices determined). These findings are of relevance both on a national level and across Africa, as some of these edible insect species are broadly consumed in Sub-Saharan (SSA) countries. Moreover, since food security was challenged due to the disruption of food systems induced by COVID-19, this study documented a needful corpus of knowledge related to commercially relevant edible insect species. Its outputs might help in implementing nutritious diets for alleviation of nutrition and food insecurity, particularly in SSA countries where entomophagy is culturally ingrained. However, results from this study also raised awareness of species-specific deficiencies, notably in leucine (for the five selected species of Lepidoptera) or in sulphur-containing AAs (for the Orthoptera representative analyzed). These limiting AAs should be taken into account for outreach to consumers, since the real benefit derived from eating insects (or from eating any other nutrient-rich foods) is necessarily associated with a varied diet. Moreover, a wholesome and varied diet is paramount to optimally meeting human requirements for essential nutrients as recommended by the FAO/WHO. Finally, preliminary insights into the distribution patterns of AAs according to the taxa of selected insects have been given. This information should be improved with larger samples and other analytical data (e.g., on edible insects’ protein digestibility and bioavailability, as well as on anti-nutrient compounds) with respect to processing methods used to increase their shelf life. Related outputs should help in tailoring insect-based diets, in labeling of insect-based products, and their marketing among SSA countries.

## Figures and Tables

**Figure 1 insects-13-00994-f001:**
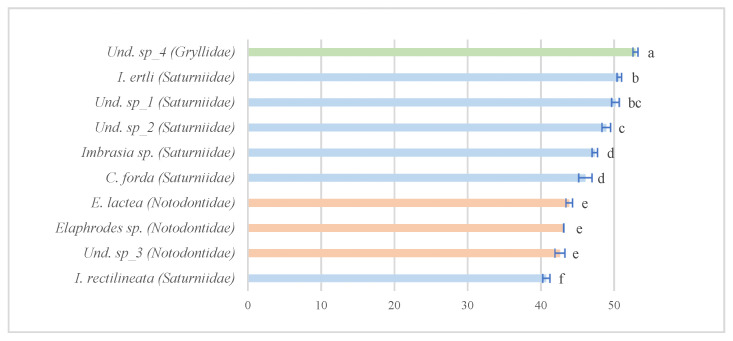
Protein content (g/100 g DM basis) of sampled edible Lepidoptera (Saturniidae or Notodontidae) and Orthoptera (Gryllidae) representatives. Different superscript letters denote a significant difference between sampled species (*p* < 0.001).

**Figure 3 insects-13-00994-f003:**
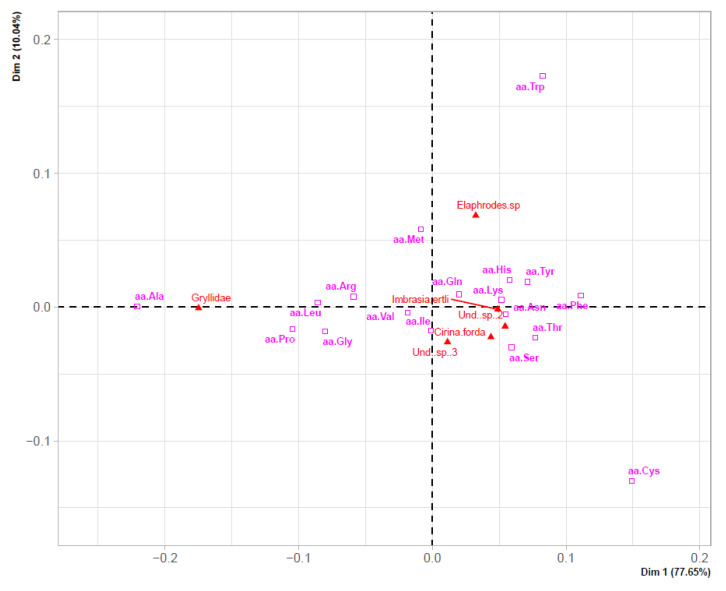
Graph of the first two principal components provided by correspondence analysis (CA) of the AA composition in the sampled edible insect species.

**Figure 4 insects-13-00994-f004:**
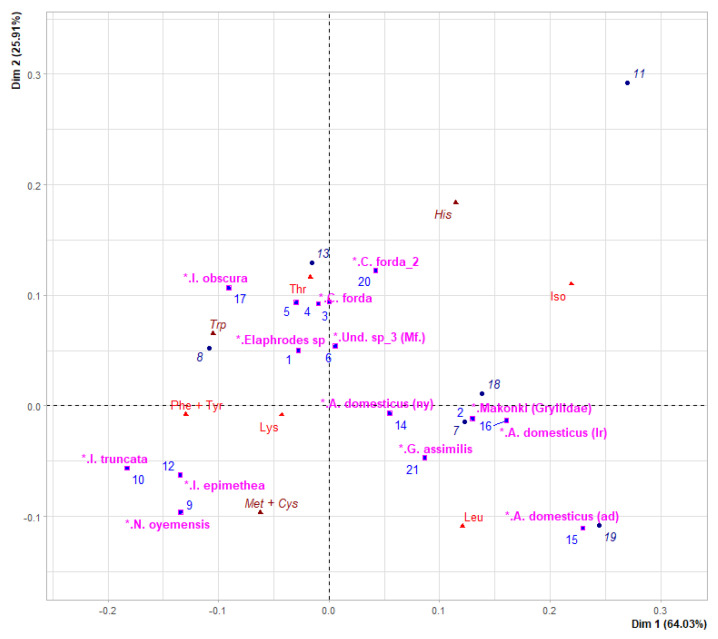
CA graph of the EAA profiles of the studied insects (from 1 to 6) and of selected species derived from the literature (from 7 to 21). Species are as follows: (1) *Elaphrodes sp*. (“Mingingi”); (2) Und. sp-4 (“Makonki”); (3) *C. forda* (“Mikwati”); (4) *I. ertli* (“Misati”); (5) Und. sp-2 (“Binkubala”); (6) Und. sp-3 (“Mifwangi-fwangi”); (7) *Gryllus bimaculatus*; (8) *Samia ricini* (pupae); (9) *N. oyemensis*; (10) *I. truncata*; (11) *I. ertli*; (12) *I. epimethea*; (13) *Anaphe venata*; (14) *A. domesticus* (nymphs); (15) *A. domesticus* (adults); (16) *A. domesticus* (larvae); (17) *I. obscura*; (18) *Gryllodes sigillatus*; (19) *Schistocerca gregaria*; (20) *C. forda*; (21) *G. assimilis*. The full names of species derived from the literature are given in Appendix A.

**Table 2 insects-13-00994-t002:** Protein content on dry matter basis for selected food sources of animal origin.

Animal Protein Sources	Description	Edible Portion (EP)	% Protein (DM) *	INFOODS _Code
Egg	Egg, chicken, local breed, raw	0.87	50.0	08_005
*Trachurus trachurus*	Atlantic horse mackerel, wild, fillet without skin, grilled (without salt or fat)	0.48	72.4	09_071
*Bos taurus*	Beef meat, lean, ca. 5% fat, grilled (without salt or fat)	1.00	78.8	07_011
*Sus domesticus*	Pork meat, moderately fat, ca. 20% fat, grilled (without salt or fat)	1.00	43.9	07_058
*G. gallus domesticus*	Chicken, light meat with skin, grilled (without salt or fat)	1.00	69.7	07_038

* The protein content of dry matter basis (DM) was calculated from data provided on fresh matter basis for an edible portion (EP) coefficient 1 (from as purchased to as described) obtained from the FAO [38].

## Data Availability

The data presented in this study are available in the article. Photographies of the studied edible species are available upon request to the corresponding author.

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
