# Peer review of "Protein Content and Amino Acid Profiles of Selected Edible Insect Species from the Democratic Republic of Congo Relevant for Transboundary Trade across Africa"

_insects, 2022, doi:10.3390/insects13110994_

Round 1
Reviewer 1 Report
Manuscript Review
insects-1954103; Title: Protein content and amino-acid profiles of selected edible insect species from the Democratic Republic of Congo relevant for transboundary trade across Africa
General comments: This manuscript brings relevant analysis on the nutritional properties, specifically related to the amino acid and protein profiles of ten edible insect species randomly collected in the DRC. It is an interesting subject that deserves more attention because of the high nutritional characteristics of the insects as food and also because of the dramatic increase of the human populations. I am not an expert in protein or amino acid quantification or characterization nevertheless, I could understand the general idea of the study which I consider relevant, well prepared and presented.
Specific comments: A few specific comments are listed below related to small corrections that must be carried out.
Before using the acronym DRC the authors must refer in full “Domenican Republic of Congo”
The font size of the last three words in the abstract must be corrected
AAS amino acid contents is presented in different ways in the simple summary and in the abstract and in the main text. Please choose AAa or AA. Please check all the acronyms used in the hole manuscript.
In the sentence ending as …”mineral and other bio-functional compounds [10,11].” The word “ mineral” should be in plural as the other compounds are mentioned.
Introduction
In the last paragraph, replace “amino acids profiles” by AAs
In the sentence” Its results might contribute;” replace the semicolon by comma
M & M
Lines 220-225 must be included in the M&M
Legends- Acronyms should not be used in the legends of tables and figs since the legends must be self-explanatory. Please check and correct table 3 legend and all others in the manuscript. In case the authors decide to use acronyms in the legends they must be followed by the full names in order to make the legends self-explanatory.
Discussion
Despite the nice results obtained with the AAs and proteins analysis explaining the importance of the insects as food sources for several countries in Africa, it would be interesting mentioning in more details that in some cases the insects might present anti nutritional properties, just to make the subject broadly and well framed.
Reviewer 2 Report
Review
Protein content and amino-acid profiles of selected edible insect species from the Democratic Republic of Congo relevant for transboundary trade across Africa
P.M. Nsevolo and the authors describe the protein content and amino acid composition (AA) of 10 edible insect species in the Democratic Republic of Congo (DRC). The study provides new insights into the nutritional potential of some insect species found in DRC. The results of the study suggest that some of the insects studied could be widely used as a complementary food source for humans.
General concept comments
It is not specified in the manuscript to which category the insect species studied belong- eggs, larvae, pupae or adults. Some insects can be consumed at different life stages, and even within an insect group, the nutritional value can vary considerably depending on the stage of metamorphosis, the origin of the insect, and its diet.
It is also not specified how the studied insects purchased from the market were processed or preserved. The section on Sample collection and identification states that some were raw, some sun-dried, and some were smoked. It is known that nutritional value is affected by the processing/preservation method.
In addition, insects may contain significant amounts of chitin, which may affect their digestibility. In the present study, only the protein content and amino acid composition were determined. For a food to be recommended as a diet, digestibility, and availability must be evaluated in addition to nutritional quality.
The possible allergenicity of insect proteins from the insects studied was also not mentioned. Allergic reactions to various insect proteins and cross-reactivity with crustacean and inhalant allergens have already been described.
It should be emphasized that further research on food safety and the effects of processing techniques on nutritional content or digestibility of edible insects is needed before insect-based diet is recommended.
English language: Sentences are too long and therefore less readable and understandable. Styling and word choice should be improved. The manuscript needs English proofreading.
Specific comments
Simple Summary:
Line 12: shelf life (not live)
Line 18: Lepidoptera contained high values in essential AAs, supporting the use of? insects for dietary supplementation
Line 20-23: The meaning of thjs sentence is unclear: Furthermore, the study also reported 3 groups (clusters) based on the AAs pro-files of species: the AAs profiles varied according to insects’ taxa, some of the studied species (namely, family Notodontidae) containing both the lowest values in severalAAs and Essential Amino Acid Index (which is a rapid calculation to determine proein nutritional quality), as compared to Saturniidae and Gryllidae.
Line 26: economic, social and ecological proceeds is written in different letters font
Abstract:
In the Abstract, the methods used in the study should be specified.
Line 29: DM-the abbreviation should be explained the first time that is written
Introduction
Line 51-56: This sentence is too long: With particular reference to developing countries in Africa facing numerous challenges (viz. fragility of food systems, overcrowding of cities, ineffective policies, poverty) [2, 5-6], sustainable and nutrient-rich foods should be consequently considered in order to mitigate food insecurity, and this all the more so as the focus is gradually shifting from the fatal effects of the COVID-19 to the threat it poses on the production and daily supply of food for children, smallholders and vulnerable populations especially [7,8].
Line 82-84: The meaninig of this sentence is unclear: This should also be highlighted for enhancing transboundary commerce of edible insects that might boost local production of species linked to potential economic, social and ecological proceeds? for rural and vulnerable communities involved in insect value chain.
Line 83: transboundary commerce? Cross-border trade
Materials and Methods
In the Materials and Methods section the two methods used for determination of proteins and AA profiles should be explained in more detail to assure the reproducibility of results.
Sample collection and identification
Since insects can be consumed as eggs, larvae, pupae or adults you must specify which category the studied insects belong to.
Protein content analysis: you must specify if the analysis was done on fresh, sun-dried or smoked samples.
Total and sulfur amino acid analysis: the HPLC analysis should be explained in more detail (standards, flow rate,..)
Line 127: 750 μM screen? filter
Line 128: each sample was (not were)
Line 130: 24 h (not H)
Line 130: opened Schott what kind (bottles, vials,…)?
Line 131: Indicate the initial pH!
Table 1: the abbreviation N/A is not explained
Line 173: you didn’t specify how the edible portion of the sampled insect species was determined. Was the whole insect used in the analysis? Was the edible portion the same for all insect species studied?
Results
In the Results section you present data in Table 2 that are not from your study but from another source- FAO. Table 2 should not be included in the Results section of the manuscript, but only cited or referred to (in the introduction or discussion).
Line 189: varied less (not few)
Line 192: spacing between words is to long
Line 194: first three (not three first)
Line 197: Furthermore, the protein content of selected edible species was compared to the protein content of most common animal-based…(You compared data not Figure)
Line 227: Ile (not Iso?)
Line 228: Tyr or Trp (not Try). Also: I. ertli didn’t have the highest content of Tyr
Line 229: had the highest Lys content (instead of indicated the highest)
Line 243: Tyr (not Try)
Discussion
Table 4 should be in Results section not Discussion
Figure 3 should be in Results section not Discussion
Lines 286 and 288: word spacing is too long
Lines 309 and 310: namely instead of viz.
Page 10, after line 349 where does this sentence belongs to: PDCAAS stands for Protein Digestibility Corrected Amino Acid Score. It is the method of choice for dietary protein quality assessment hitherto
Line 361: word spacing is to long
Line 366: variations in protein content of food can also be due to different processing methods of food!
Conclusion and perspectives
Line 361: word spacing is to long
Line 394: namely instead of viz.
Line 395: word spacing is to long
Line 400: have (not has) been given
References
Line 491: Ref. 32: spacing
Reviewer 3 Report
In the present manuscript, authors investigated the protein quality of the insect species commercially available in the DRC. The topic fits in the journal scope and the manuscript is well written. However, major revisions are needed in terms of study design and in-depth investigation and discussion of the findings. In particular:
- the manuscript reported only the results about the amino acids profile and protein content of 10 different insects, which in my opinion does not represent enough material for providing an original article. I suggest authors to perform additional analysis (proximate composition, fatty acid profile) in order to provide a deep investigation in the nutritional profile of these commercial insect species.
- Authors reported in the abstract that 10 different insect species have been characterized for their amino acids (AA) profile, however in the result section, only 6 species are reported. I suggest authors to include also the AA profile of missing species.
- please provide more information (e.g. mL of acid/base, time of hydrolysis, temperature) about the analysis performed for tryptophan and sulphurated AA analysis, in order to make it reproducible by other scientists
- I suggest authors to harmonize the name used for identifying the different insects along the manuscript, in order to help future readers during the results comparison. As an example, in line 187 authors referred to lepidopterans reported in Figure 1, even if in the figure is not specified which insects belonged to this order.
- authors should enrich the discussion of their findings with the results about these insect species which are already published in literature. Furthermore, it would be interesting to provide a comparative study with the AA profile of insect species commercially available in other part of the world.
Round 2
Reviewer 2 Report
All the reviewers comments have been adressed. I believe the manuscript has been greatly improved.
Reviewer 3 Report
The manuscript has been revised by taking into account the suggestion proposed by reviewer.
I have really appreciate the Correspondence Analysis for highlighting the similarities and differences among literature data.
Just two small remarks:
- In the abstract you have corrected to "six" the number of insect species analysed for both protein content and AA profile. However, 10 species have been analysed for total protein and only six for the AA profile. I suggest authors to include this specification also in the abstract.
- In 3.2 I suggest to specify the species of insects which have been selected for the AA characterisation, prior to present the results
